# Decoupled Variational Gaussian Inference

**Mohammad Emtiyaz Khan**
Ecole Polytechnique Fédérale de Lausanne (EPFL), Switzerland
emtiyaz@gmail.com

## Abstract

Variational Gaussian (VG) inference methods that optimize a lower bound to the marginal likelihood are a popular approach for Bayesian inference. A difficulty remains in computation of the lower bound when the latent dimensionality $L$ is large. Even though the lower bound is concave for many models, its computation requires optimization over $O(L^2)$ variational parameters. Efficient reparameterization schemes can reduce the number of parameters, but give inaccurate solutions or destroy concavity leading to slow convergence. We propose decoupled variational inference that brings the best of both worlds together. First, it maximizes a Lagrangian of the lower bound reducing the number of parameters to $O(N)$, where $N$ is the number of data examples. The reparameterization obtained is unique and recovers maxima of the lower-bound even when it is not concave. Second, our method maximizes the lower bound using a sequence of convex problems, each of which is parallellizable over data examples. Each gradient computation reduces to prediction in a *pseudo* linear regression model, thereby avoiding all direct computations of the covariance and only requiring its linear projections. Theoretically, our method converges at the same rate as existing methods in the case of concave lower bounds, while remaining convergent at a reasonable rate for the non-concave case.

## 1 Introduction

Large-scale Bayesian inference remains intractable for many models, such as logistic regression, sparse linear models, or dynamical systems with non-Gaussian observations. Approximate Bayesian inference requires fast, robust, and reliable algorithms. In this context, algorithms based on variational Gaussian (VG) approximations are growing in popularity [17, 3, 13, 6] since they strike a favorable balance between accuracy, generality, speed, and ease of use.

VG inference remains problematic for models with large latent-dimensionality. While some variants are convex [3], they require $O(L^2)$ variational parameters to be optimized, where $L$ is the latent-dimensionality. This slows down the optimization. One solution is to restrict the covariance representations by naive mean-field [2] or restricted Cholesky [3], but this can result in considerable loss of accuracy when significant posterior correlations exist. An alternative is to reparameterize the covariance to obtain $O(N)$ number of parameters, where $N$ is the number of data examples [17]. However, this destroys the convexity and converges slowly [12]. A recent approach called dual variational inference [10] obtains fast convergence while retaining this parameterization, but is applicable to only some models such as Poisson regression.

In this paper, we propose an approach called decoupled variational Gaussian inference which extends the dual variational inference to a large class of models. Our method relies on the theory of Lagrangian multiplier methods. While remaining widely applicable, our approach reduces the number of variational parameters similar to [17, 10] and converges at similar convergence rates as convex methods such as [3]. Our method is similar in spirit to parallel expectation-propagation (EP) but has provable convergence guarantees even when likelihoods are not log-concave.

## 2 The Model

In this paper, we apply our method for Bayesian inference on Latent Gaussian Models (LGMs). This choice is motivated by a large amount of existing work on VG approximations for LGMs [16, 17, 3, 10, 12, 11, 7, 2], and because LGMs include many popular models, such as Gaussian processes, Bayesian regression and classification, Gaussian Markov random field, and probabilistic PCA. An extensive list of these models is given in Chapter 1 of [9]. We have also included few examples in the supplementary material.

Given a vector of observations $\mathbf{y}$ of length $N$, LGMs model the dependencies among its components using a latent Gaussian vector $\mathbf{z}$ of length $L$. The joint distribution is shown below.

$$p(\mathbf{y}, \mathbf{z}) = \prod_{n=1}^{N} p_n(y_n|\eta_n)p(\mathbf{z}), \qquad \boldsymbol{\eta} = \mathbf{W}\mathbf{z}, \qquad p(\mathbf{z}) := \mathcal{N}(\mathbf{z}|\boldsymbol{\mu}, \boldsymbol{\Sigma}) \tag{1}$$

where $\mathbf{W}$ is a known real-valued matrix of size $N \times L$, and is used to define linear predictors $\boldsymbol{\eta}$. Each $\eta_n$ is used to model the observation $y_n$ using a link function $p_n(y_n|\eta_n)$. The exact form of this function depends on the type of observations, e.g. a Bernoulli-logit distribution can be used for binary data [14, 7]. See the supplementary material for an example. Usually, an exponential family distribution is used, although there are other choices (such as T-distribution [8, 17]). The parameter set $\boldsymbol{\theta}$ includes $\{\mathbf{W}, \boldsymbol{\mu}, \boldsymbol{\Sigma}\}$ and other parameters of the link function and is assumed to be known. We suppress $\boldsymbol{\theta}$ in our notation, for simplicity.

In Bayesian inference, we wish to compute expectations with respect to the posterior distribution $p(\mathbf{z}|\mathbf{y})$ which is shown below. Another important task is the computation of the marginal likelihood $p(\mathbf{y})$ which can be maximized to estimate parameters $\boldsymbol{\theta}$, for example, using empirical Bayes [18].

$$p(\mathbf{z}|\mathbf{y}) \propto \prod_{n=1}^{N} p(y_n|\eta_n)\mathcal{N}(\mathbf{z}|\boldsymbol{\mu}, \boldsymbol{\Sigma}) \quad , \quad p(\mathbf{y}) = \int \prod_{n=1}^{N} p(y_n|\eta_n)\mathcal{N}(\mathbf{z}|\boldsymbol{\mu}, \boldsymbol{\Sigma})\, d\mathbf{z} \tag{2}$$

For non-Gaussian likelihoods, both of these tasks are intractable. Applications in practice demand good approximations that scale favorably in $N$ and $L$.

## 3 VG Inference by Lower Bound Maximization

In variational Gaussian (VG) inference [17], we assume the posterior to be a Gaussian $q(\mathbf{z}) = \mathcal{N}(\mathbf{z}|\mathbf{m}, \mathbf{V})$. The posterior mean $\mathbf{m}$ and covariance $\mathbf{V}$ form the set of variational parameters, and are chosen to maximize the variational lower bound to the log marginal likelihood shown in Eq. (3). To get this lower bound, we first multiply and divide by $q(\mathbf{z})$ and then apply Jensen's inequality using the concavity of $\log$.

$$\log p(\mathbf{y}) = \log \int q(\mathbf{z}) \frac{\prod_n p(y_n|\eta_n)p(\mathbf{z})}{q(\mathbf{z})}\, d\mathbf{z} \geq \mathbb{E}_{q(z)}\left[\log \frac{\prod_n p(y_n|\eta_n)p(\mathbf{z})}{q(\mathbf{z})}\right] \tag{3}$$

The simplified lower bound is shown in Eq. (4). The detailed derivation can be found in Eqs. (4)–(7) in [11] (and in the supplementary material). Below, we provide a summary of its components.

$$\max_{\mathbf{m}, \mathbf{V}\succ 0} -\mathbb{D}[q(\mathbf{z}) \,\|\, p(\mathbf{z})] - \sum_{n=1}^{N} f_n(\bar{m}_n, \bar{\sigma}_n), \quad f_n(\bar{m}_n, \bar{\sigma}_n) := \mathbb{E}_{\mathcal{N}(\eta_n|\bar{m}_n, \bar{\sigma}_n^2)}[-\log p(y_n|\eta_n)] \tag{4}$$

The first term is the KL-divergence $\mathbb{D}[q \,\|\, p] = \mathbb{E}_q[\log q(\mathbf{z}) - \log p(\mathbf{z})]$, which is jointly concave in $(\mathbf{m}, \mathbf{V})$. The second term sums over data examples, where each term denoted by $f_n$ is the expectation of $-\log p(y_n|\eta_n)$ with respect to $\eta_n$. Since $\eta_n = \mathbf{w}_n^T\mathbf{z}$, it follows a Gaussian distribution $q(\eta_n) = \mathcal{N}(\bar{m}_n, \bar{\sigma}_n^2)$ with mean $\bar{m}_n = \mathbf{w}_n^T\mathbf{m}$ and variances $\bar{\sigma}_n^2 = \mathbf{w}_n^T\mathbf{V}\mathbf{w}_n$. The terms $f_n$ are not always available in closed form, but can be computed using quadrature or look-up tables [14]. Note that unlike many other methods such [2, 11, 10, 7, 21], we do not bound or approximate these terms. Such approximations lead to loss of accuracy.

We denote the lower bound of Eq. (3) by $f$ and expand it below in Eq. (5):

$$f(\mathbf{m}, \mathbf{V}) := \tfrac{1}{2}[\log |\mathbf{V}| - \mathrm{Tr}(\mathbf{V}\boldsymbol{\Sigma}^{-1}) - (\mathbf{m} - \boldsymbol{\mu})^T\boldsymbol{\Sigma}^{-1}(\mathbf{m} - \boldsymbol{\mu}) + L] - \sum_{n=1}^{N} f_n(\bar{m}_n, \bar{\sigma}_n) \tag{5}$$

Here $|\mathbf{V}|$ denotes the determinant of $\mathbf{V}$. We now discuss existing methods and their pros and cons.

## 3.1 Related Work

A straight-forward approach is to optimize Eq. (5) directly in $(\mathbf{m}, \mathbf{V})$ [2, 3, 14, 11]. In practice, direct methods are slow and memory-intensive because of the very large number $L + L(L+1)/2$ of variables. Challis and Barber [3] show that for log-concave likelihoods $p(y_n|\eta_n)$, the original problem Eq. (4) is jointly concave in $\mathbf{m}$ and the Cholesky factor of $\mathbf{V}$. This fact, however, does not result in any reduction in the number of parameters, and they propose to use factorizations of a restricted form, which negatively affects the approximation accuracy.

[17] and [16] note that the optimal $\mathbf{V}^*$ must be of the form $\mathbf{V}^* = [\boldsymbol{\Sigma}^{-1} + \mathbf{W}^T \text{diag}(\boldsymbol{\lambda})\mathbf{W}]^{-1}$, which suggests reparameterizing Eq. (5) in terms of $L+N$ parameters $(\mathbf{m}, \boldsymbol{\lambda})$, where $\boldsymbol{\lambda}$ is the new variable. However, the problem is not concave in this alternative parameterization [12]. Moreover, as shown in [12] and [10], convergence can be exceedingly slow. The coordinate-ascent approach of [12] and dual variational inference [10] both speed-up convergence, but only for a limited class of models.

A range of different deterministic inference approximations exist as well. The local variational method is convex for log-concave potentials and can be solved at very large scales [23], but applies only to models with super-Gaussian likelihoods. The bound it maximizes is provably less tight than Eq. (4) [22, 3] making it less accurate. Expectation propagation (EP) [15, 21] is more general and can be more accurate than most other approximations mentioned here. However, it is based on a saddle-point rather than an optimization problem, and the standard EP algorithm does not always converge and can be numerically unstable. Among these alternatives, the variational Gaussian approximation stands out as a compromise between accuracy and good algorithmic properties.

## 4 Decoupled Variational Gaussian Inference using a Lagrangian

We simplify the form of the objective function by *decoupling* the KL divergence term from the terms including $f_n$. In other words, we separate the prior distribution from the likelihoods. We do so by introducing real-valued auxiliary-variables $h_n$ and $\sigma_n > 0$, such that the following constraints hold: $h_n = \bar{m}_n$ and $\sigma_n = \bar{\sigma}_n$. This gives us the following (equivalent) optimization problem over $\mathbf{x} := \{\mathbf{m}, \mathbf{V}, \mathbf{h}, \boldsymbol{\sigma}\}$,

$$\max_{\mathbf{x}} g(\mathbf{x}) := \tfrac{1}{2}\left[\log|\mathbf{V}| - \text{Tr}(\mathbf{V}\boldsymbol{\Sigma}^{-1}) - (\mathbf{m} - \boldsymbol{\mu})^T \boldsymbol{\Sigma}^{-1}(\mathbf{m} - \boldsymbol{\mu}) + L\right] - \sum_{n=1}^{N} f_n(h_n, \sigma_n) \quad (6)$$

subject to constraints $c_n^1(\mathbf{x}) := h_n - \mathbf{w}_n^T\mathbf{m} = 0$ and $c_n^2(\mathbf{x}) := \tfrac{1}{2}(\sigma_n^2 - \mathbf{w}_n^T\mathbf{V}\mathbf{w}_n) = 0$ for all $n$.

For log-concave likelihoods, the function $g(\mathbf{x})$ is concave in $\mathbf{V}$, unlike the original function $f$ (see Eq. (5)) which is concave with respect to Cholesky of $\mathbf{V}$. The difficulty now lies with the non-linear constraints $c_n^2(\mathbf{x})$. We will now establish that the new problem gives rise to a convenient parameterization, but does not affect the maximum.

The significance of this reformulation lies in its Lagrangian, shown below.

$$\mathcal{L}(\mathbf{x}, \boldsymbol{\alpha}, \boldsymbol{\lambda}) := g(\mathbf{x}) + \sum_{n=1}^{N} \alpha_n(h_n - \mathbf{w}_n^T\mathbf{m}) + \tfrac{1}{2}\lambda_n(\sigma_n^2 - \mathbf{w}_n^T\mathbf{V}\mathbf{w}_n) \quad (7)$$

Here, $\alpha_n, \lambda_n$ are Lagrangian multipliers for the constraints $c_n^1(\mathbf{x})$ and $c^2(\mathbf{x})$. We will now show that the maximum of $f$ of Eq. (5) can be parameterized in terms of these multipliers, and that this reparameterization is unique. The following theorem states this result along with three other useful relationships between the maximum of Eq. (5), (6), and (7). Proof is in the supplementary material.

**Theorem 4.1.** *The following holds for maxima of Eq. (5), (6), and (7):*

1. *A stationary point $\mathbf{x}^*$ of Eq. (6) will also be a stationary point of Eq. (5). For every such stationary point $\mathbf{x}^*$, there exist unique $\boldsymbol{\alpha}^*$ and $\boldsymbol{\lambda}^*$ such that,*

$$\mathbf{V}^* = [\boldsymbol{\Sigma}^{-1} + \mathbf{W}^T diag(\boldsymbol{\lambda}^*)\mathbf{W}]^{-1}, \quad \mathbf{m}^* = \boldsymbol{\mu} - \boldsymbol{\Sigma}\mathbf{W}^T\boldsymbol{\alpha}^* \quad (8)$$

2. *The $\alpha_n^*$ and $\lambda_n^*$ depend on the gradient of function $f_n$ and satisfy the following conditions,*

$$\nabla_{h_n} f_n(h_n^*, \sigma_n^*) = \alpha_n^*, \quad \nabla_{\sigma_n} f_n(h_n^*, \sigma_n^*) = \sigma_n^*\lambda_n^* \quad (9)$$

*where $h_n^* = \mathbf{w}_n^T \mathbf{m}^*$ and $(\sigma_n^*)^2 = \mathbf{w}_n^T \mathbf{V}^* \mathbf{w}_n$ for all $n$ and $\bigtriangledown_x f(x^*)$ denotes the gradient of $f(x)$ with respect to $x$ at $x = x^*$.*

3. *When $\{\mathbf{m}^*, \mathbf{V}^*\}$ is a local maximizer of Eq. (5), then the set $\{\mathbf{m}^*, \mathbf{V}^*, \mathbf{h}^*, \boldsymbol{\sigma}^*, \boldsymbol{\alpha}^*, \boldsymbol{\lambda}^*\}$ is a strict maximizer of Eq. (7).*

4. *When likelihoods $p(y_n|\eta_n)$ are log-concave, there is only one global maximum of $f$, and any $\{\mathbf{m}^*, \mathbf{V}^*\}$ obtained by maximizing Eq. (7) will be the global maximizer of Eq. (5).*

Part 1 establishes the parameterization of $(\mathbf{m}^*, \mathbf{V}^*)$ by $(\boldsymbol{\alpha}^*, \boldsymbol{\lambda}^*)$ and its uniqueness, while part 2 shows the conditions that $(\boldsymbol{\alpha}^*, \boldsymbol{\lambda}^*)$ satisfy. This form has also been used in [12] for Gaussian Processes where a fixed-point iteration was employed to search for $\boldsymbol{\lambda}^*$. Part 3 shows that such parameterization can be obtained at maxima of the Lagrangian rather than minima or saddle-points. The final part considers the case when $f$ is concave and shows that the global maximum can be obtained by maximizing the Lagrangian. Note that concavity of the lower bound is required for the last part only and the other three parts are true irrespective of concavity.

Detailed proof of the theorem is given in the supplementary material.

Note that the conditions of Eq. (9) restrict the values that $\alpha_n^*$ and $\lambda_n^*$ can take. Their values will be valid only in the range of the gradients of $f_n$. This is unlike the formulation of [17] which does not constrain these variables, but is similar to the method of [10]. We will see later that our algorithm makes the problem infeasible for values outside this range. Ranges of these variables vary depending on the likelihood $p(y_n|\eta_n)$. However, we show below in Eq. (10) that $\lambda_n^*$ is always strictly positive for log-concave likelihoods. The first equality is obtained using Eq. (9), while the second equality is simply change of variables from $\sigma_n$ to $\sigma_n^2$. The third equality is obtained using Eq. (19) from [17]. The final inequality is obtained since $f_n$ is convex for all log-concave likelihoods ($\bigtriangledown_{xx} f(x)$ denotes the Hessian of $f(x)$).

$$\lambda_n^* = \sigma_n^{*-1} \bigtriangledown_{\sigma_n} f_n(h_n^*, \sigma_n^*) = 2 \bigtriangledown_{\sigma_n^2} f_n(h_n^*, \sigma_n^*) = \bigtriangledown_{h_n h_n}^2 f_n(h_n^*, \sigma_n^*) > 0 \qquad (10)$$

## 5 Optimization Algorithms for Decoupled Variational Gaussian Inference

Theorem 4.1 suggests that the optimal solution can be obtained by maximizing $g(\mathbf{x})$ or the Lagrangian $\mathcal{L}$. The maximization is difficult for two reasons. First, the constraints $c_n^2(\mathbf{x})$ are non-linear and second the function $g(\mathbf{x})$ may not always be concave. Note that it is not easy to apply the augmented Lagrangian method or first-order methods (see Chapter 4 of [1]) because their application would require storage of $\mathbf{V}$. Instead, we use a method based on linearization of the constraints which will avoid explicit computation and storage of $\mathbf{V}$. First, we will show that when $g(\mathbf{x})$ is concave, we can maximize it by minimizing a sequence of convex problems. We will then solve each convex problem using the dual-variational method of [10].

### 5.1 Linearly Constrained Lagrangian (LCL) Method

We now derive an algorithm based on the linearly constrained Lagrangian (LCL) method [19]. The LCL approach involves linearization of the non-linear constraints and is an effective method for large-scale optimization, e.g. in packages such as MINOS [24]. There are variants of this method that are globally convergent and robust [4], but we use the variant described in Chapter 17 of [24].

**The final algorithm:** See Algorithm 1. We start with a $\boldsymbol{\alpha}, \boldsymbol{\lambda}$ and $\boldsymbol{\sigma}$. At every iteration $k$, we *minimize* the following dual:

$$\min_{\boldsymbol{\alpha}, \boldsymbol{\lambda} \in \mathcal{S}} -\tfrac{1}{2} \log |\boldsymbol{\Sigma}^{-1} + \mathbf{W}^T \operatorname{diag}(\boldsymbol{\lambda}) \mathbf{W}| + \tfrac{1}{2} \boldsymbol{\alpha}^T \widetilde{\boldsymbol{\Sigma}} \boldsymbol{\alpha} - \widetilde{\boldsymbol{\mu}}^T \boldsymbol{\alpha} + \sum_{n=1}^{N} f_n^{k*}(\alpha_n, \lambda_n) \qquad (11)$$

Here, $\widetilde{\boldsymbol{\Sigma}} = \mathbf{W} \boldsymbol{\Sigma} \mathbf{W}^T$ and $\widetilde{\boldsymbol{\mu}} = \mathbf{W} \boldsymbol{\mu}$. The functions $f_n^{k*}$ are obtained as follows:

$$f_n^{k*}(\alpha_n, \lambda_n) := \max_{h_n, \sigma_n > 0} -f_n(h_n, \sigma_n) + \alpha_n h_n + \tfrac{1}{2} \lambda_n \sigma_n^k (2\sigma_n - \sigma_n^k) - \tfrac{1}{2} \lambda_n^k (\sigma_n - \sigma_n^k)^2 \qquad (12)$$

where $\lambda_n^k$ and $\sigma_n^k$ were obtained at the previous iteration.

**Algorithm 1** Linearly constrained Lagrangian (LCL) method for VG approximation

---
Initialize $\boldsymbol{\alpha}, \boldsymbol{\lambda} \in \mathcal{S}$ and $\boldsymbol{\sigma} \succ 0$.
**for** $k = 1, 2, 3, \ldots$ **do**
   $\boldsymbol{\lambda}^k \leftarrow \boldsymbol{\lambda}$ and $\boldsymbol{\sigma}^k \leftarrow \boldsymbol{\sigma}$.
   **repeat**
      For all $n$, compute predictive mean $\hat{m}_n^*$ and variances $\hat{v}_n^*$ using linear regression (Eq. (13))
      For all $n$, in parallel, compute $(h_n^*, \sigma_n^*)$ that maximizes Eq. (12).
      Find next $(\boldsymbol{\alpha}, \boldsymbol{\lambda}) \in \mathcal{S}$ using gradients $g_n^\alpha = h_n^* - \hat{m}_n^*$ and $g_n^\lambda = \frac{1}{2}[-(\sigma_n^k)^2 + 2\sigma_n^k \sigma_n - \hat{v}_n^*]$.
   **until** convergence
**end for**

---

The constraint set $\mathcal{S}$ is a box constraints on $\alpha_n$ and $\lambda_n$ such that a global minimum of Eq. (12) exists. We will show some examples later in this section.

**Efficient gradient computation:** An advantage of this approach is that the gradient at each iteration can be computed efficiently, especially for large $N$ and $L$. The gradient computation is decoupled into two terms. The first term can be computed by computing $f_n^{k*}$ in parallel, while the second term involves prediction in a linear model. The gradients with respect to $\alpha_n$ and $\lambda_n$ (derived in the supplementary material) are given as $g_n^\alpha := h_n^* - \hat{m}_n^*$ and $g_n^\lambda := \frac{1}{2}[-(\sigma_n^k)^2 + 2\sigma_n^k \sigma_n^* - \hat{v}_n^*]$, where $(h_n^*, \sigma_n^*)$ are maximizers of Eq. (12) and $\hat{v}_n^*$ and $\hat{m}_n^*$ are computed as follows:

$$\hat{v}_n^* := \mathbf{w}_n^T \mathbf{V}_n^* \mathbf{w}_n = \mathbf{w}_n^T (\boldsymbol{\Sigma}^{-1} + \mathbf{W}^T \mathrm{diag}(\boldsymbol{\lambda}) \mathbf{W})^{-1} \mathbf{w}_n = \widetilde{\Sigma}_{nn} - \widetilde{\boldsymbol{\Sigma}}_{n,:} (\widetilde{\boldsymbol{\Sigma}} + \mathrm{diag}(\boldsymbol{\lambda})^{-1})^{-1} \widetilde{\boldsymbol{\Sigma}}_{n,:}$$

$$\hat{m}_n^* := \mathbf{w}_n^T \mathbf{m}_n^* = \mathbf{w}_n^T (\boldsymbol{\mu} - \boldsymbol{\Sigma} \mathbf{W}^T \boldsymbol{\alpha}) = \widetilde{\mu}_n - \widetilde{\boldsymbol{\Sigma}}_{n,:} \boldsymbol{\alpha} \tag{13}$$

The quantities $(h_n^*, \sigma_n^*)$ can be computed in parallel over all $n$. Sometimes, this can be done in closed form (as we shown in the next section), otherwise we can compute them by numerically optimizing over two-dimensional functions. Since these problems are only two-dimensional, a Newton method can be easily implemented to obtain fast convergence.

The other two terms $\hat{v}_n^*$ and $\hat{m}_n^*$ can be interpreted as predictive means and variances of a *pseudo* linear model, e.g. compare Eq. (13) with Eq. 2.25 and 2.26 of Rasmussen's book [18]. Hence every gradient computation can be expressed as Bayesian prediction in a linear model for which we can use existing implementation. For example, for binary or multi-class GP classification, we can reuse efficient implementation of GP regression. In general, *we can use a Bayesian inference in a conjuate model to compute the gradient of a non-conjugate model*. This way the method also avoids forming $\mathbf{V}^*$ and work only with its linear projections which can be efficiently computed using vector-matrix-vector products.

The "decoupling" nature of our algorithm should now be clear. The non-linear computations depending on the data, are done in parallel to compute $h_n^*$ and $\sigma_n^*$. These are completely decoupled from linear computations for $\hat{m}_n$ and $\hat{v}_n$. This is summarized in Algorithm (1).

**Derivation:** To derive the algorithm, we first linearize the constraints. Given multiplier $\boldsymbol{\lambda}^k$ and a point $\mathbf{x}^k$ at the $k$'th iteration, we linearize the constraints $c_n^2(\mathbf{x})$:

$$\bar{c}_{nk}^2(\mathbf{x}) := c_n^2(\mathbf{x}^k) + \bigtriangledown c_n^2(\mathbf{x}^k)^T (\mathbf{x} - \mathbf{x}^k) \tag{14}$$

$$= \tfrac{1}{2}[(\sigma_n^k)^2 - \mathbf{w}_n^T \mathbf{V}^k \mathbf{w}_n + 2\sigma_n^k(\sigma_n - \sigma_n^k) - (\mathbf{w}_n^T \mathbf{V} \mathbf{w}_n - \mathbf{w}_n^T \mathbf{V}^k \mathbf{w}_n)] \tag{15}$$

$$= -\tfrac{1}{2}[(\sigma_n^k)^2 - 2\sigma_n^k \sigma_n + \mathbf{w}_n^T \mathbf{V} \mathbf{w}_n] \tag{16}$$

Since we want the linearized constraint $\bar{c}_{nk}^2(\mathbf{x})$ to be close to the original constraint $c_n^2(\mathbf{x})$, we will penalize the difference between the two.

$$c_n^2(\mathbf{x}) - \bar{c}_{nk}^2(\mathbf{x}) = \tfrac{1}{2}\{\sigma_n^2 - \mathbf{w}_n^T \mathbf{V} \mathbf{w}_n - [-(\sigma_n^k)^2 + 2\sigma_n^k \sigma_n - \mathbf{w}_n^T \mathbf{V} \mathbf{w}_n]\} = \tfrac{1}{2}(\sigma_n - \sigma_n^k)^2 \tag{17}$$

The key point is that this term is independent of $\mathbf{V}$, allowing us to obtain a closed-form solution for $\mathbf{V}^*$. This will also be crucial for the extension to non-concave case in the next section.

The new $k$'th subproblem is defined with the linearized constraints and the penalization term:

$$\max_{\mathbf{x}} g^k(\mathbf{x}) := g(\mathbf{x}) - \sum_{n=1}^{N} \tfrac{1}{2}\lambda_n^k (\sigma_n - \sigma_n^k)^2 \tag{18}$$

$$s.t. \quad h_n - \mathbf{w}_n^T \mathbf{m}_n = 0 \quad, \quad -\tfrac{1}{2}[(\sigma_n^k)^2 - 2\sigma_n^k \sigma_n + \mathbf{w}_n^T \mathbf{V} \mathbf{w}_n] = 0, \quad \forall n$$

This is a concave problem with linear constraints and can be optimized using dual variational inference [10]. Detailed derivation is given in the supplementary material.

**Convergence:** When LCL algorithm converges, it has quadratic convergence rates [19]. However, it may not always converge. Globally convergent methods do exist (e.g. [4]) although we do not explore them in this paper. Below, we present a simple approach that improves the convergence for non log-concave likelihoods.

**Augmented Lagrangian Methods for non log-concave likelihoods:** When the likelihood $p(y_n|\eta_n)$ are not log-concave, the lower bound can contain local minimum, making the optimization difficult for function $f(\mathbf{m}, \mathbf{V})$. In such scenarios, the algorithm may not converge for all starting values.

The convergence of our approach can be improved for such cases. We simply add an augmented Lagrangian term $[\bar{c}_{nk}^2(\mathbf{x})]^2$ to the linearly constrained Lagrangian defined in Eq. (18), as shown below [24]. Here, $\delta_i^k > 0$ and $i$ is the $i$'th iteration of $k$'th subproblem:

$$g_{aug}^k(\mathbf{x}) := g(\mathbf{x}) - \sum_{n=1}^{N} \tfrac{1}{2}\lambda_n^k (\sigma_n - \sigma_n^k)^2 + \tfrac{1}{2}\delta_i^k (\sigma_n - \sigma_n^k)^4 \tag{19}$$

subject to the same constraints as Eq. (18).

The sequence $\delta_i^k$ can either be set to a constant or be increased slowly to ensure convergence to a local maximum. More details on setting this sequence and its affect on the convergence can be found in Chapter 4.2 of [1]. It is in fact possible to know the value of $\delta_i^k$ such that the algorithm always converge. This value can be set by examining the *primal function* - a function with respect to the *deviations* in constraints. It turns out that it should be set larger than the largest eigenvalues of the Hessian of the primal function at 0. A good discussion of this can be found in Chapter 4.2 of [1].

The fact that that the linearized constraint $\bar{c}_{nk}^2(\mathbf{x})$ does not depend on $\mathbf{V}$ is very useful here since addition of this term then only affects computation of $f_n^{k*}$. We modify the algorithm by simply changing the computation to optimization of the following function:

$$\max_{h_n, \sigma_n > 0} -f_n(h_n, \sigma_n) + \alpha_n h_n + \tfrac{1}{2}\lambda_n \sigma_n^k (2\sigma_n - \sigma_n^k) - \tfrac{1}{2}\lambda_n^k (\sigma_n - \sigma_n^k)^2 - \frac{\delta_i^k}{2}(\sigma_n - \sigma_n^k)^4 \tag{20}$$

It is clear from this that the augmented Lagrangian term is trying to "convexify" the non-convex function $f_n$, leading to improved convergence.

**Computation of $f_n^{k*}(\alpha, \lambda_n)$** These functions are obtained by solving the optimization problem shown in Eq. (12). In some cases, we can compute these functions in closed form. For example, as shown in the supplementary material, we can compute $h^*$ and $\sigma^*$ in closed form for Poisson likelihood as shown below. We also show the range of $\alpha_n$ and $\lambda_n$ for which $f_n^{k*}$ is finite.

$$\sigma_n^* = \frac{\lambda_n + \lambda_n^k}{y_n + \alpha_n + \lambda_n^k}\sigma_n^k, \quad h_n^* = -\tfrac{1}{2}\sigma_n^{*2} + \log(y_n + \alpha_n), \quad \mathcal{S} = \{\alpha_n > -y_n, \lambda_n > 0, \forall n\} \tag{21}$$

An expression for Laplace likelihood is also derived in the supplementary material.

When we do not have a closed-form expression for $f_n^{k*}$, we can use a 2-D Newton method for optimization. To facilitate convergence, we must warm-start the optimization. When $f_n$ is concave, this usually converges in few iterations, and since we can parallelize over $n$, a significant speed-up can be obtained. A significant engineering effort is required for parallelization and for our experiments in this paper, we have not done so.

An issue that remains open is the evaluation of the range $\mathcal{S}$ for which each $f_n^{k*}$ is finite. For now, we have simply set it to the range of gradients of function $f_n$ as shown by Eq. 9 (also see the last paragraph in that section). It is not clear whether this will always assure convergence for the 2-D optimization.

**Prediction:** Given $\boldsymbol{\alpha}^*$ and $\boldsymbol{\lambda}^*$, we can compute the predictions by using equations similar to GP regression. See details in Rasmussen's book [18].

# 6 Results

We demonstrate the advantages of our approach on a binary GP classification problem. We model the binary data using Bernoulli-logit likelihoods. Function $f_n$ are computed to a reasonable accuracy using the piecewise bound [14] with 20 pieces.

We apply this model to a subproblem of the USPS digit data [18]. Here, the task is to classify between 3's vs. 5's. There are a total of 1540 data examples with feature dimensionality of 256. Since we want to compare the convergence, we will show results for different data sizes by subsampling randomly from these examples.

We set $\boldsymbol{\mu} = 0$ and use a squared-exponential kernel, for which the $(i, j)$th entry of $\boldsymbol{\Sigma}$ is defined as: $\boldsymbol{\Sigma}_{ij} = -\sigma^2 \exp[-\frac{1}{2}||\mathbf{x}_i - \mathbf{x}_j||^2/s]$ where $\mathbf{x}_i$ is $i$'th feature. We show results for $\log(\sigma) = 4$ and $\log(s) = -1$ which corresponds to a difficult case where VG approximations converge slowly (due to the ill-conditioning of the Kernel) [18]. Our conclusions hold for other parameter settings as well.

We compare our algorithm with the approach of Opper and Archambeau [17] and Challis and Barber [3]. We refer to them as 'Opper' and 'Cholesky', respectively. We call our approach 'Decoupled'. For all methods, we use L-BFGS method for optimization (implemented in `minFunc` by Mark Schmidt), since a Newton method would be too expensive for large $N$. All algorithms were stopped when the subsequent changes in the lower bound value of Eq. 5 were less than $10^{-4}$. All methods were randomly initialized. Our results are not sensitive to initialization. We compare convergence in terms of the value of lower bound. The prediction errors show very similar trend, therefore we do not present them.

The results are summarized in Figure 1. Each plot shows the negative of the lower bound vs time in seconds for increasing data sizes $N = 200, 500, 1000$ and $1500$. For Opper and Cholesky, we show markers for every iteration. For decoupled, we show markers after completion of each subproblem. We can not see the result of first subproblem here, and the first visible marker is obtained from the second subproblem onwards.

We see that as the data size increases, Decoupled converges faster than the other methods, showing a clear advantage over other methods for large dimensionality.

# 7 Discussion and Future Work

In this paper, we proposed the decoupled VG inference method for approximate Bayesian inference. We obtain efficient reparameterization using a Lagrangian to the lower bound. We showed that such a parameterization is unique, even for non log-concave likelihood functions, and the maximum of the lower bound can be obtained by maximizing the Lagrangian. For concave likelihood function, our method recovers the global maximum. We proposed a linearly constrained Lagrangian method to maximize the Lagrangian. The algorithm has the desired property that it reduces each gradient computation to a linear model computation, while parallelizing non-linear computations over data examples. Our proposed algorithm is capable of attaining convergence rates similar to convex methods.

Unlike methods such as mean-field approximation, our method preserves all posterior correlations and can be useful towards generalizing stochastic variational inference (SVI) methods [5] to nonconjugate models. Existing SVI methods rely on mean-field approximations and are widely applied for conjugate models. Under our method, we can stochastically include only few constraints to maximize the Lagrangian. This amounts to a low-rank approximation of the covariance matrix and can be used to construct an unbiased estimate of the gradient.

We have focused only on latent Gaussian models for simplicity. It is easy to extend our approach to other non-Gaussian latent models, e.g. sparse Bayesian linear model [21] and Bayesian nonnegative matrix factorization [20]. Similar decoupling method can also be applied to general latent variable models. Note that a choice of proper posterior distribution is required to get an efficient parameterization of the posterior.

It is also possible to get sparse posterior covariance approximation using our decoupled formulation. One possible idea is to use Hinge type of loss to approximate the likelihood terms. Using the dualization similar to what we have shown here would give us a sparse posterior covariance.

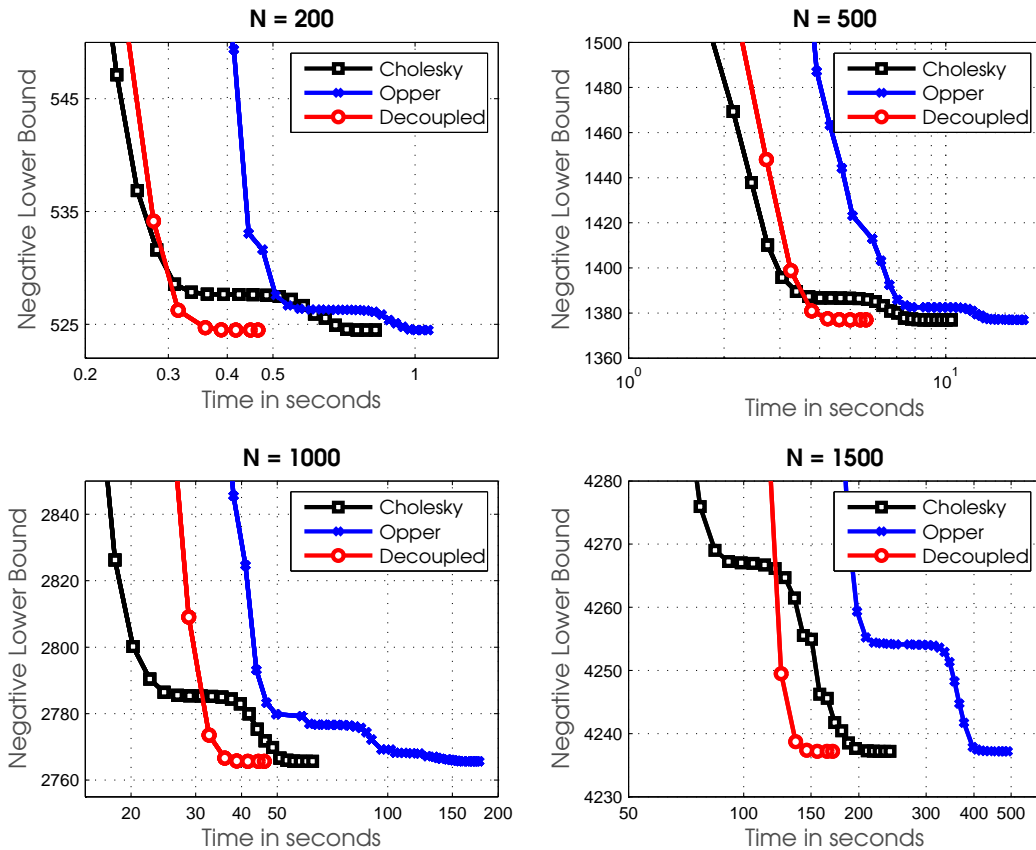

Figure 1: Convergence results for a GP classification on the USPS-3vs5 data set. Each plot shows the negative of the lower bound vs time in seconds for data sizes $N = 200, 500, 1000$ and 1500. For Opper and Cholesky, we show markers for every iteration. For decoupled, we show markers after completion of each subproblem. We can not see the result of first subproblem here, and the first visible marker is obtained from the second subproblem. As the data size increases, Decoupled converges faster, showing a clear advantage over other methods for large dimensionality.

A weakness of our paper is a lack of strong experiments showing that the decoupled method indeed converge at a fast rate. The implementation of decoupled method requires a good engineering effort for it to scale to big data. In future, we plan to have an efficient implementation of this method and demonstrate that this enables variational inference to scale to large data.

**Acknowledgments**

This work was supported by School of Computer Science and Communication at EPFL. I would specifically like to thank Matthias Grossglauser, Rudiger Urbanke, and Jame Larus for providing me support and funding during this work. I would like to personally thank Volkan Cevher, Quoc Tran-Dinh, and Matthias Seeger from EPFL for early discussions of this work and Marc Desgroseilliers from EPFL for checkin some proofs.

I would also like to thank the reviewers for their valuable feedback. The experiments in this paper are less extensive than what I promised them. Due to time and space constraints, I have not been able to add all of them. More experiments will appear in an arXiv version of this paper.

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
