[Supplementary Material]

# Supplementary Material for "Decoupled Variational Gaussian Inference"

**Mohammad Emtiyaz Khan**
Ecole Polytechnique Fédérale de Lausanne (EPFL), Switzerland
emtiyaz@gmail.com

## A  Latent Gaussian models

Below, we show a graphical model for LGMs and GPs. For GPs, $N$ is the number of data examples, $\mathbf{z}$ is the latent function therefore latent dimensionality $L = N$. $\boldsymbol{\mu}$ and $\boldsymbol{\Sigma}$ are the mean function vector and covariance function matrix respectively. In this case the covariance matrix of $\mathbf{z}$ will be of size $N^2$.

Similarly, for Bayesian logistic regression, $N$ is the number of examples pairs with $y_n$ as labels while $\mathbf{x}_n$ as features. $L$ is the feature dimensionality (length of features $\mathbf{x}_n$). The latent vector $\mathbf{z}$ contains regression weights and $\boldsymbol{\mu}$ and $\boldsymbol{\Sigma}$ are prior mean and covariance for $\mathbf{z}$.

(a) LGM　　　　　　　　　(b) GP

## B  Derivation of the lower bound of Eq. (4) and (5)

We can obtain the lower bound as shown below:

$$\log p(\mathbf{y}) \geq \mathbb{E}_{q(z)}\left[\log \frac{\prod_n p(y_n|\eta_n)p(\mathbf{z})}{q(\mathbf{z})}\right] = \mathbb{E}_{q(z)}\left[\log \frac{p(\mathbf{z})}{q(\mathbf{z})}\right] + \sum_n \mathbb{E}_{q(z)}[\log p(y_n|\eta_n)] \quad (1)$$

$$= -\mathbb{D}[q(\mathbf{z}) \,\|\, p(\mathbf{z})] + \sum_n \mathbb{E}_{\mathcal{N}(\eta_n|\bar{m}_n,\bar{\sigma}_n^2)}[\log p(y_n|\eta_n)] \quad (2)$$

where the last step is obtained by using the fact that $\eta_n = \mathbf{w}_n^T \mathbf{z}$ and therefore the expectation can be written in terms of the distribution of $\eta_n$ which is Gaussian with mean $\bar{m}_n = \mathbf{w}_n^T \mathbf{m}$ and variance $\bar{\sigma}_n^2 = \mathbf{w}_n^T \mathbf{V} \mathbf{w}_n$. Expanding the KL-divergence terms give us the desired lower bound.

## C Proof of theorem 4.1

We make use of the first-order necessary and second-order sufficient conditions of Lagrangian multiplier theory (see Chapter 3 of Bertsekas's book on "Nonlinear Programming"). There are two pre-requisites for these conditions to apply. The first requirement is that $g(\mathbf{x})$ and the constraints $c_n^1(\mathbf{x})$ and $c_n^2(\mathbf{x})$ are twice-differentiable, which clearlyholds in our case.

The second requirement is that a maximizer $\mathbf{x}^*$ of $g(\mathbf{x})$ is regular. A maximizer $\mathbf{x}^*$ is regular when the gradients of constraints are linearly independent. Since all of our constraints operate on different subset of variables, they are linearly independent. Hence both conditions hold.

**Proof of first two parts:** The first two parts can be proved by simply taking the gradient of the Lagrangian with respect to $\mathbf{x}$, $\boldsymbol{\alpha}$ and $\boldsymbol{\lambda}$. We first take the gradient of the Lagrangian (of Eq. (7) in the paper) with respect to $\mathbf{m}$ and $\mathbf{V}$.

$$\nabla_m \mathcal{L} = -\boldsymbol{\Sigma}^{-1}(\mathbf{m} - \boldsymbol{\mu}) - \sum_n \alpha_n \mathbf{w}_n \tag{3}$$

$$\nabla_V \mathcal{L} = \tfrac{1}{2}\mathbf{V}^{-1} - \tfrac{1}{2}\boldsymbol{\Sigma}^{-1} - \tfrac{1}{2}\sum_n \lambda_n \mathbf{w}_n \mathbf{w}_n^T \tag{4}$$

Equating to zero, we get Eq. (8) in the paper.

Similarly, taking gradient wrt $h_n$ and $\sigma_n$, we get the following:

$$\nabla_{h_n} \mathcal{L} = -\nabla_{h_n} f_n(h_n, \sigma_n) + \alpha_n \tag{5}$$
$$\nabla_{\sigma_n} \mathcal{L} = -\nabla_{\sigma_n} f_n(h_n, \sigma_n) + \sigma_n \lambda_n \tag{6}$$

Equating to zero gives us the Eq. (9) in the paper.

Taking the gradient wrt $\alpha_n$ and $\lambda_n$ gives us the condition $h_n - \mathbf{w}_n^T \mathbf{m}_n = 0$ and $\sigma_n - \mathbf{w}_n^T \mathbf{V} \mathbf{w}_n = 0$.

This gives us all the conditions given in the first two parts of the theorem. Proposition 3.1.1 of Bertsekas book ensures the uniqueness of $\boldsymbol{\alpha}^*$ and $\boldsymbol{\lambda}^*$ and also that this will be a stationary point of $g(\mathbf{x})$.

What remains to prove is whether a pair $(\mathbf{m}^*, \mathbf{V}^*)$ obtained from vector $\mathbf{x}^*$ will satisfy the first-order condition for $f$ or not (i.e. Eq. 5 in the paper). This can be checked by taking the gradient of $f$, and then simply substituting Eq. (8) of the paper in the resulting expression. We show this for $\mathbf{V}$ below in Eq. 7. The first equality shows the gradient of $f$ with respect to $\mathbf{V}$. Here, $\mathbf{g}^*$ is a vector containing gradients of $f_n$ with respect to $(\bar{\sigma}_n^*)^2 := \mathbf{w}_n^T \mathbf{V}^* \mathbf{w}_n$. The second equality is obtained after substituting Eq. (8) of the paper into the gradient and using Eq. (9) of the paper which states that $\lambda_n^* = 2 \nabla_{(\sigma_n)^2} f_n^*$ so that $\boldsymbol{\lambda}^* = 2\mathbf{g}^*$.

$$\nabla_V f(\mathbf{m}^*, \mathbf{V}^*) = (\mathbf{V}^*)^{-1} - \boldsymbol{\Sigma}^{-1} - 2\mathbf{W}^T \text{diag}(\mathbf{g}^*)\mathbf{W} \tag{7}$$
$$= \boldsymbol{\Sigma}^{-1} + \mathbf{W}^T \boldsymbol{\Lambda}^* \mathbf{W} - \boldsymbol{\Sigma}^{-1} - \mathbf{W}^T \text{diag}(\boldsymbol{\lambda}^*)\mathbf{W} \tag{8}$$
$$= 0 \tag{9}$$

Similarly, we can prove the condition for $\mathbf{m}$ too.

**Proof of the third part:** Proof for this part is tedious. We fist give an outline the proof here and then go into details. At a local maximizer of $f$, the Hessian should be negative-definite by definition. We also know from the first two parts of the Theorem that the corresponding set $\{\mathbf{m}^*, \mathbf{V}^*, \mathbf{h}^*, \boldsymbol{\sigma}^*, \boldsymbol{\alpha}^*, \boldsymbol{\lambda}^*\}$ satisfies the first-order condition. To prove that this set will also be a strict maximizer, we use the Proposition 3.2.1 from Bertsekas book. According to the proposition, we only need to show that the Hessian of Lagrangian is negative-definite in the tangent plane of the constraints.

We first demonstrate this for a 1-D case, since it is easier to gain intuition why the theorem is true. We also assume for this part that the function $f_n$ are convex. Consider the following 1-D function (with $\boldsymbol{\mu} = 0$):

$$\mathcal{L}(m, v, h, \sigma) := \tfrac{1}{2}[\log v - (v + m^2)/\sigma^2] - f_1(h, \sigma) + \alpha(h - m) + \tfrac{1}{2}\lambda(\sigma^2 - v) \tag{10}$$

The tangent plane at $\mathbf{x}^*$ consists of all $\mathbf{x}$ such that its inner product with the gradient of constraints is 0. The constraints are $h - m = 0$ and $(\sigma^2 - v)/2 = 0$, and the gradients with respect to $\mathbf{x}$ is

summarized below, in the left by $2 \times 4$ matrix. The two rows are gradients of the two constraints wrt to $[m, v, h, \sigma]$ at $[m^*, v^*, h^*, \sigma^*]$. Null space is defined by all vectors for which the inner product with each row is 0.

$$\begin{bmatrix} -1 & 0 & 1 & 0 \\ 0 & -1/2 & 0 & \sigma^* \end{bmatrix} \begin{bmatrix} m \\ v \\ h \\ \sigma \end{bmatrix} = \begin{bmatrix} 0 \\ 0 \end{bmatrix} \tag{11}$$

This gives us the tangent space $\{\mathbf{x} : m = h, v = 2\sigma^*\sigma\}$.

The Hessian with respect to $\mathbf{x}^* = \{m^*, v^*, h^*, \sigma^*\}$ is a diagonal matrix (here $f_1^*$ is $f_1(h^*, \sigma^*)$, not the conjugate as defined in the main paper).

$$\mathbf{H}^* = \begin{bmatrix} -\frac{1}{\sigma^2} & 0 & 0 & 0 \\ 0 & -\frac{1}{2(v^*)^2} & 0 & 0 \\ 0 & 0 & -\frac{\partial^2 f_1^*}{\partial h^2} & 0 \\ 0 & 0 & 0 & -\frac{\partial^2 f_1^*}{\partial \sigma^2} + \lambda^* \end{bmatrix} \tag{12}$$

We now show that $\mathbf{x}^T \mathbf{H}^* \mathbf{x} < 0$ for all $\mathbf{x}$ in the tangent space with $\mathbf{H}^*$ being the Hessian at $\mathbf{x}^*$.

$$\mathbf{x}^T \mathbf{H}^* \mathbf{x} = -\frac{m^2}{\sigma^2} - \frac{v^2}{2(v^*)^2} - h^2 \frac{\partial^2 f_1^*}{\partial h^2} - \sigma^2 \left( \frac{\partial^2 f_1^*}{\partial \sigma^2} - \lambda^* \right) \tag{13}$$

The first and third terms are -ve for sure, so we focus on the second and fourth terms only. We get Eq. 15 using the fact that in tangent space $v = 2\sigma^*\sigma$, i.e. $\sigma = v/(2\sigma^*)$. Then we use that at optimum $(\sigma^*)^2 = v^*$ to get Eq. 16. Next Equation is obtained simply by taking $v^2/4v^*$ out. After this we substitute $\lambda^* = 1/v^* - 1/\sigma^2$, which gives us the Eq. 18 and 19, which is strictly negative due to convexity of $f_1$.

$$-\frac{v^2}{2(v^*)^2} - \sigma^2 \left( \frac{\partial^2 f_1^*}{\partial \sigma^2} - \lambda^* \right) \tag{14}$$

$$= -\frac{v^2}{2(v^*)^2} - \frac{v^2}{4(\sigma^*)^2} \left( \frac{\partial^2 f_1^*}{\partial \sigma^2} - \lambda^* \right) \tag{15}$$

$$= -\frac{v^2}{2(v^*)^2} - \frac{v^2}{4v^*} \left( \frac{\partial^2 f_1^*}{\partial \sigma^2} - \lambda^* \right) \tag{16}$$

$$= -\frac{v^2}{4v^*} \left( \frac{2}{v^*} - \lambda^* + \frac{\partial^2 f_1^*}{\partial \sigma^2} \right) \tag{17}$$

$$= -\frac{v^2}{4v^*} \left( \frac{2}{v^*} - \frac{1}{v^*} + \frac{1}{\sigma^2} + \frac{\partial^2 f_1^*}{\partial \sigma^2} \right) \tag{18}$$

$$= -\frac{v^2}{4v^*} \left( \frac{1}{v^*} + \frac{1}{\sigma^2} + \frac{\partial^2 f_1^*}{\partial \sigma^2} \right) \tag{19}$$

$$< 0 \tag{20}$$

This proves the required result.

Generalization of this proof to multivariate case is tedious, since we have to deal with second derivative with respect to the matrix $\mathbf{V}$. However, similar to 1-D case, the Hessian will be a block-diagonal matrix, so that $\mathbf{x}^T \mathbf{H}^* \mathbf{x}$ will have four terms, each corresponding to $\mathbf{m}, \mathbf{V}, \mathbf{h}, \boldsymbol{\sigma}$. Positivity of first and third term can be shown in a similar way, so we don't consider them. We only show positivity of second and fourth terms which depend on $\mathbf{V}$ and $\boldsymbol{\sigma}$. So, in the following, we ignore $\mathbf{m}$ and $\mathbf{h}$.

We can avoid explicitly writing the second derivative wrt to the matrix, by using the fact that if $(\mathbf{m}^*, \mathbf{V}^*)$ are maximizers of $f$, they will satisfy the second-order condition for function $f(\mathbf{x})$, i.e. the Hessian of $f$ will be negative-definiteness at the maximizer. We will now formally express this relationship and then use it to show a similar expression for the Lagrangian.

Consider the objective function $f$ after omitting the terms depending on $\mathbf{m}$ and $\mathbf{h}$.

$$\tfrac{1}{2} [\log |\mathbf{V}| - \operatorname{Tr}(\mathbf{V}\boldsymbol{\Sigma}^{-1})] - \sum_{n=1}^{N} f_n(\bar{\sigma}_n) \tag{21}$$

Since $(\mathbf{m}^*, \mathbf{V}^*)$ is a maximizer, it will satisfy the second-order condition. Taking the derivative of $f$ twice (Eq. (5) in the paper) and denoting $\mathbf{v} = \text{vec}(\mathbf{V})$ and the second-derivative of $\log|\mathbf{V}| + \text{Tr}(\mathbf{V}\boldsymbol{\Sigma}^{-1})$ wrt $\mathbf{v}$ by $\mathbf{H}_d$, the condition can be written as follows,

$$\mathbf{v}^T\mathbf{H}_d^*\mathbf{v} - \sum_{n=1}^{N}\sum_{k,l,r,s} V_{kl}V_{rs}\frac{\partial}{\partial V_{kl}}\frac{\partial}{\partial V_{rs}}f_n(\bar{\sigma}_n^*) < 0 \tag{22}$$

where $\bar{\sigma}_n^* = \sqrt{\mathbf{w}_n^T\mathbf{V}^*\mathbf{w}_n}$. We will now express the second term in terms of $\bar{\sigma}_n^*$ and the derivative wrt $\bar{\sigma}_n$. Using $\partial\bar{\sigma}_n/\partial V_{kl} = W_{nk}W_{nl}/(2\bar{\sigma}_n)$, we can write the following:

$$\frac{\partial}{\partial V_{rs}}\frac{\partial}{\partial V_{kl}}f_n(\bar{\sigma}_n^*) \tag{23}$$

$$= \frac{\partial}{\partial V_{rs}}\left[W_{nk}W_{nl}\frac{1}{2\bar{\sigma}_n^*}\frac{\partial f_n(\bar{\sigma}_n^*)}{\partial\bar{\sigma}_n}\right] \tag{24}$$

$$= \tfrac{1}{2}W_{nk}W_{nl}\left[\frac{1}{\bar{\sigma}_n^*}\frac{\partial^2 f_n(\bar{\sigma}_n^*)}{\partial V_{rs}\partial\bar{\sigma}_n} + \frac{\partial f_n(\bar{\sigma}_n^*)}{\partial\bar{\sigma}_n}\frac{\partial(\bar{\sigma}_n^*)^{-1}}{\partial V_{kl}}\right] \tag{25}$$

$$= \tfrac{1}{2}W_{nk}W_{nl}\left[W_{nr}W_{ns}\frac{1}{2(\bar{\sigma}_n^*)^2}\frac{\partial^2 f_n(\bar{\sigma}_n^*)}{\partial\bar{\sigma}_n^2} - W_{nl}W_{ns}\frac{1}{2(\bar{\sigma}_n^*)^3}\frac{\partial f_n(\bar{\sigma}_n^*)}{\partial\bar{\sigma}_n}\right] \tag{26}$$

$$= W_{nk}W_{nl}W_{nr}W_{ns}\frac{1}{(2\bar{\sigma}_n^*)^2}\left[\frac{\partial^2 f_n(\bar{\sigma}_n^*)}{\partial\bar{\sigma}_n^2} - \frac{1}{\bar{\sigma}_n^*}\frac{\partial f_n(\bar{\sigma}_n^*)}{\partial\bar{\sigma}_n}\right] \tag{27}$$

Hence, we can rewrite the second term of Eq. 22 as follows,

$$\mathbf{v}^T\mathbf{H}_d^*\mathbf{v} < \sum_{n=1}^{N}\sum_{k,l,r,s} V_{kl}V_{rs}\frac{\partial}{\partial V_{kl}}\frac{\partial}{\partial V_{rs}}f_n(\bar{\sigma}_n^*) \tag{28}$$

$$= \sum_{n=1}^{N}\sum_{k,l,r,s} V_{kl}V_{rs}W_{nk}W_{nl}W_{nr}W_{ns}\frac{1}{(2\bar{\sigma}_n^*)^2}\left[\frac{\partial^2 f_n(\bar{\sigma}_n^*)}{\partial\bar{\sigma}_n^2} - \frac{1}{\bar{\sigma}_n^*}\frac{\partial f_n(\bar{\sigma}_n^*)}{\partial\bar{\sigma}_n}\right] \tag{29}$$

$$= \sum_{n=1}^{N}\frac{(\mathbf{w}_n^T\mathbf{V}\mathbf{w}_n)^2}{(2\bar{\sigma}_n^*)^2}\left[\frac{\partial^2 f_n(\bar{\sigma}_n^*)}{\partial\bar{\sigma}_n^2} - \frac{1}{\bar{\sigma}_n^*}\frac{\partial f_n(\bar{\sigma}_n^*)}{\partial\bar{\sigma}_n}\right] \tag{30}$$

We will now use this relation to show that the Hessian of the Lagrangian is negative-definite in the tangent plane.

Taking second derivative of the Lagrangian of Eq. (6) in the paper wrt $\mathbf{V}$ and $\boldsymbol{\sigma}$, we get the first equation. For the second equation, we substitute Eq. 30. For the third equation, we use the relation that $\sigma_n^*\lambda_n^* = \partial f_n(\sigma_n^*)/\partial\sigma_n$ and $\sigma_n = \bar{\sigma}_n^*$, and also that in the tangent plane, the following holds: $\{\mathbf{x} : 2\sigma_n^*\sigma_n = \mathbf{x}_n^T\mathbf{V}\mathbf{x}_n, \forall n\}$. Using these the third equation simplifies to 0.

$$\mathbf{x}^T\mathbf{H}^*\mathbf{x} = \mathbf{v}^T\mathbf{H}_d^*\mathbf{v} - \sum_{n=1}^{N}\sigma_n^2\left[\frac{\partial^2 f_n(\sigma_n^*)}{\partial\sigma_n^2} - \lambda_n^*\right] \tag{31}$$

$$< \sum_{n=1}^{N}\frac{(\mathbf{x}_n^T\mathbf{V}\mathbf{x}_n)^2}{(2\bar{\sigma}_n^*)^2}\left[\frac{\partial^2 f_n(\bar{\sigma}_n^*)}{\partial\bar{\sigma}_n^2} - \frac{1}{\bar{\sigma}_n^*}\frac{\partial f_n(\bar{\sigma}_n^*)}{\partial\bar{\sigma}_n}\right] - \sum_{n=1}^{N}\sigma_n^2\left[\frac{\partial^2 f_n(\sigma_n^*)}{\partial\sigma_n^2} - \lambda_n^*\right] \tag{32}$$

$$= \sum_{n=1}^{N}\frac{(2\sigma_n^*\sigma_n)^2}{(2\sigma_n^*)^2}\left[\frac{\partial^2 f_n(\bar{\sigma}_n^*)}{\partial\bar{\sigma}_n^2} - \frac{1}{\bar{\sigma}_n^*}\frac{\partial f_n(\bar{\sigma}_n^*)}{\partial\bar{\sigma}_n}\right] - \sum_{n=1}^{N}\sigma_n^2\left[\frac{\partial^2 f_n(\sigma_n^*)}{\partial\sigma_n^2} - \frac{1}{\sigma_n^*}\frac{\partial f_n(\sigma_n^*)}{\partial\sigma_n}\right] \tag{33}$$

$$= 0 \tag{34}$$

which proves the required result.

**Proof of the last part:** For the last part, we make use of the result of Challis and Barber (2011) which establishes the strong-concavity of $f$ with respect to the Cholesky $\mathbf{L}$ of $\mathbf{V}$ for log-concave likelihoods. Since the mapping from $\mathbf{L}$ to $\mathbf{V}$ is one-to-one, it follows that there will be a unique maximum at $\mathbf{V}^* = \mathbf{L}^*(\mathbf{L}^*)^T$ when $\mathbf{L}^*$ is the global maximizer of $f$. Since $\mathbf{L}$ is a positive-definite matrix, we can conclude that the gradient with respect to $\mathbf{L}$ is zero if and only if the gradient with respect to $\mathbf{V}$ is zero. Therefore, any maximizer of $\mathcal{L}$ will be a global maximizer of $f$.

# D Derivation of the dual

The new $k$'th subproblem is defined as follows:

$$\max_{\mathbf{x}} g^k(\mathbf{x}) := g(\mathbf{x}) - \tfrac{1}{2}\sum_{n=1}^{N}\lambda_n^k(\sigma_n - \sigma_n^k)^2 \tag{35}$$

$$\text{s.t.} \quad h_n - \mathbf{w}_n^T\mathbf{m}_n = 0, \quad -\tfrac{1}{2}[(\sigma_n^k)^2 - 2\sigma_n^k\sigma_n + \mathbf{w}_n^T\mathbf{V}\mathbf{w}_n] = 0, \quad \forall n$$

We can expand $g(\mathbf{x})$ using the definition of Eq. (6) of the paper and write the Lagrangian as shown below,

$$\tfrac{1}{2}\left[\log|\mathbf{V}| - \text{Tr}(\mathbf{V}\mathbf{\Sigma}^{-1}) - (\mathbf{m}-\boldsymbol{\mu})^T\mathbf{\Sigma}^{-1}(\mathbf{m}-\boldsymbol{\mu}) + L\right] - \sum_{n=1}^{N}f_n(h_n,\sigma_n) - \tfrac{1}{2}\sum_{n=1}^{N}\lambda_n^k(\sigma_n - \sigma_n^k)^2$$

$$+ \sum_{n=1}^{N}\alpha_n(h_n - \mathbf{w}_n^T\mathbf{m}_n) - \tfrac{1}{2}\lambda_n[(\sigma_n^k)^2 - 2\sigma_n^k\sigma_n + \mathbf{w}_n^T\mathbf{V}\mathbf{w}_n] \tag{36}$$

Next, we rearrange the terms. We move all terms containing $\mathbf{m}$ and $\mathbf{V}$ in the first line, while the ones containing $\mathbf{h}$ and $\boldsymbol{\sigma}$ to the second line.

$$\tfrac{1}{2}\left[\log|\mathbf{V}| - \text{Tr}(\mathbf{V}\mathbf{\Sigma}^{-1}) - (\mathbf{m}-\boldsymbol{\mu})^T\mathbf{\Sigma}^{-1}(\mathbf{m}-\boldsymbol{\mu}) + L\right] - \sum_{n=1}^{N}\alpha_n\mathbf{w}_n^T\mathbf{m}_n + \tfrac{1}{2}\lambda_n\mathbf{w}_n^T\mathbf{V}\mathbf{w}_n$$

$$+ \sum_{n=1}^{N} -f_n(h_n,\sigma_n) + \alpha_n h_n - \tfrac{1}{2}\lambda_n[(\sigma_n^k)^2 - 2\sigma_n^k\sigma_n] - \tfrac{1}{2}\lambda_n^k(\sigma_n - \sigma_n^k)^2 \tag{37}$$

We can now differentiate w.r.t $\mathbf{m}$ and $\mathbf{V}$ and get the following closed-form solution.

$$\mathbf{V}^* = [\mathbf{\Sigma}^{-1} + \mathbf{W}^T\text{diag}(\boldsymbol{\lambda})\mathbf{W}]^{-1}, \quad \mathbf{m}^* = \boldsymbol{\mu} - \mathbf{\Sigma}\mathbf{W}^T\boldsymbol{\alpha} \tag{38}$$

Similarly, $h_n^*$ and $\sigma_n^*$ can be obtained by optimizing the second line which gives us the $f_n^{k*}(\alpha_n, \lambda_n)$.

$$f_n^{k*}(\alpha_n, \lambda_n) := \max_{h_n,\sigma_n > 0} -f_n(h_n,\sigma_n) + \alpha_n h_n + \tfrac{1}{2}\lambda_n\sigma_n^k(2\sigma_n - \sigma_n^k) - \tfrac{1}{2}\lambda_n^k(\sigma_n - \sigma_n^k)^2 \tag{39}$$

The second line of Eq. (37), therefore, simplifies to the function $f_n^{k*}(\alpha_n, \lambda_n)$. Note that even though this maximization involves a concave function, existence of the global maximum depends on values of $\boldsymbol{\alpha}$ and $\boldsymbol{\lambda}$. Denoting all the values of $\boldsymbol{\alpha}$ and $\boldsymbol{\lambda}$ such that a global minimum of Eq. (47) exist by set $\mathcal{S}$. We will optimize the dual over this set $\mathcal{S}$.

Substituting this back in the first line of Eq. (37), we can simplify the first line as shown below. The first two lines are simply a rearrangement of terms. In third line, we substitute the value of $\mathbf{V}^*$ and $\mathbf{m}^*$. In fourth line, we simplify since some terms cancel out.

$$\tfrac{1}{2}\left[\log|\mathbf{V}^*| - \text{Tr}(\mathbf{V}^*\mathbf{\Sigma}^{-1}) - (\mathbf{m}^*-\boldsymbol{\mu})^T\mathbf{\Sigma}^{-1}(\mathbf{m}^*-\boldsymbol{\mu}) + L\right] - \sum_{n=1}^{N}\alpha_n\mathbf{w}_n^T\mathbf{m}_n^* + \tfrac{1}{2}\lambda_n\mathbf{w}_n^T\mathbf{V}^*\mathbf{w}_n$$

$$= \tfrac{1}{2}\left[\log|\mathbf{V}^*| - \text{Tr}[\mathbf{V}^*(\mathbf{\Sigma}^{-1} + \mathbf{W}^T\text{diag}(\boldsymbol{\lambda}^*)\mathbf{W})] - (\mathbf{m}^*-\boldsymbol{\mu})^T\mathbf{\Sigma}^{-1}(\mathbf{m}^*-\boldsymbol{\mu}) + L\right] - \boldsymbol{\alpha}^T\mathbf{W}\mathbf{m}^*$$

$$= \tfrac{1}{2}\left[-\log|\mathbf{\Sigma}^{-1} + \mathbf{W}^T\text{diag}(\boldsymbol{\lambda}^*)\mathbf{W}| - L - \boldsymbol{\alpha}^T\mathbf{W}\mathbf{\Sigma}\mathbf{W}^T\boldsymbol{\alpha} + L\right] - \boldsymbol{\alpha}^T\mathbf{W}(\boldsymbol{\mu} - \mathbf{\Sigma}\mathbf{W}^T\boldsymbol{\alpha}) \tag{40}$$

$$= -\tfrac{1}{2}\log|\mathbf{\Sigma}^{-1} + \mathbf{W}^T\text{diag}(\boldsymbol{\lambda}^*)\mathbf{W}| + \tfrac{1}{2}\boldsymbol{\alpha}^T\mathbf{W}\mathbf{\Sigma}\mathbf{W}^T\boldsymbol{\alpha} - \boldsymbol{\alpha}^T\mathbf{W}\boldsymbol{\mu} \tag{41}$$

Adding the second term obtained by maximizing over $\mathbf{h}$ and $\boldsymbol{\sigma}$, we get the final dual:

$$\min_{\boldsymbol{\alpha},\boldsymbol{\lambda}\in\mathcal{S}} -\tfrac{1}{2}\log|\mathbf{\Sigma}^{-1} + \mathbf{W}^T\text{diag}(\boldsymbol{\lambda}^*)\mathbf{W}| + \tfrac{1}{2}\boldsymbol{\alpha}^T\widetilde{\mathbf{\Sigma}}\boldsymbol{\alpha} - \boldsymbol{\alpha}^T\widetilde{\boldsymbol{\mu}} + \sum_{n=1}^{N}f_n^{k*}(\alpha_n, \lambda_n) \tag{42}$$

## D.1 Gradient of the dual

Derivative of $f_n^{k*}$ wrt $\alpha_n$ is simply the value of $h_n$ at which the maximum of Eq. (47) is attained. We denote this by $h_n^*$. Similarly, derivative wrt $\lambda_n$ is equal to $\frac{1}{2}\sigma_n^k(2\sigma_n^* - \sigma_n^k)$ where $\sigma_n^*$ is the maximizer of Eq. (47).

Derivative of rest of the terms wrt $\boldsymbol{\alpha}$ is straightforward. The derivative with respect to $\lambda_n$ can be obtained as shown below using the chain rule. Denoting $\mathbf{A} := \mathbf{V}^{-1} = \boldsymbol{\Sigma}^{-1} + \mathbf{W}^T \text{diag}(\boldsymbol{\lambda}^*)\mathbf{W}$, we get the first line by using the chain rule.

$$\nabla_{\lambda_n} \log|\mathbf{A}| = \text{Tr}\left[(\nabla_A \log|\mathbf{A}|)\nabla_{\lambda_n}\mathbf{A}\right] = \text{Tr}\left[\mathbf{V}\nabla_{\lambda_n}\mathbf{A}\right] = \text{Tr}\left[\mathbf{V}\mathbf{w}_n\mathbf{w}_n^T\right] = \mathbf{w}_n^T\mathbf{V}\mathbf{w}_n = \hat{v}_n^* \tag{43}$$

Therefore,

$$g_n^\alpha := h_n^* - \hat{m}_n^* \tag{44}$$

$$g_n^\lambda := \tfrac{1}{2}[-(\sigma_n^k)^2 + 2\sigma_n^k\sigma_n^* - \hat{v}_n^*] \tag{45}$$

# E Derivation of the function $f_n^{k*}$

For some likelihoods, we can compute the function in closed form. We demonstrate it for Poisson and Laplace likelihoods.

**Poisson likelihood:** The likelihood is given as $\log p(y_n|\eta_n) = y_n\eta_n - \exp(\eta_n) + \text{cnst}$, where $y$ is a positive integer. The $f_n$ term can be computed in closed form using the identity that expectation of $\exp(\eta_n)$ wrt a Gaussian $\mathcal{N}(\eta_n|h_n,\sigma_n^2)$ is $\exp(h_n + \sigma_n^2/2)$. The expression is given below:

$$f_n(h_n,\sigma_n) := -y_nh_n + \exp\left(h_n + \tfrac{1}{2}\sigma_n^2\right) \tag{46}$$

Plugging this into the definition of $f_n^{k*}$, we get the following optimization problem:

$$f_n^{k*}(\alpha_n,\lambda_n) := \max_{h_n,\sigma_n>0} f_n^k((\alpha_n,\lambda_n,h_n,\sigma_n) \tag{47}$$

$$f_n^k(\alpha_n,\lambda_n,h_n,\sigma_n) := y_nh_n - e^{h_n+\frac{1}{2}\sigma_n^2} + \alpha_nh_n + \tfrac{1}{2}\lambda_n\sigma_n^k(2\sigma_n - \sigma_n^k) - \tfrac{1}{2}\lambda_n^k(\sigma_n - \sigma_n^k)^2 \tag{48}$$

Taking derivative wrt to $h_n$ and $\sigma_n$, we get the following:

$$\nabla_{h_n} f_n^k = y_n - e^{h_n+\frac{1}{2}\sigma_n^2} + \alpha_n \tag{49}$$

$$\nabla_{\sigma_n} f_n^k = -\sigma_n e^{h_n+\frac{1}{2}\sigma_n^2} + \lambda_n\sigma_n^k - \lambda_n^k(\sigma_n - \sigma_n^k) \tag{50}$$

Setting the first gradient to zero, and simplifying, we get the expression for $h^*$:

$$e^{h_n^*+\frac{1}{2}\sigma_n^{*2}} = y_n + \alpha_n \tag{51}$$

Plugging this in the second gradient and equating it to zero, we simplify as below to get an expression for $\sigma_n^{k*}$:

$$-\sigma_n^* e^{h_n^*+\frac{1}{2}\sigma_n^{*2}} + \lambda_n\sigma_n^k - \lambda_n^k(\sigma_n^* - \sigma_n^k) = 0 \tag{52}$$

$$\Rightarrow \quad -\sigma_n^*(y_n + \alpha_n) + \lambda_n\sigma_n^k - \lambda_n^k(\sigma_n^* - \sigma_n^k) = 0 \tag{53}$$

$$\Rightarrow \quad -\sigma_n^*(y_n + \alpha_n + \lambda_n^k) + \sigma_n^k(\lambda_n + \lambda_n^k) = 0 \tag{54}$$

$$\Rightarrow \quad \sigma_n^* = \frac{\lambda_n + \lambda_n^k}{y_n + \alpha_n + \lambda_n^k}\sigma_n^k \tag{55}$$

Plugging this in Eq. 51, we can also get the value of $h_n^*$. Below are the expressions for $h_n^*$ and $\sigma_n^*$.

$$\sigma_n^* = \frac{\lambda_n + \lambda_n^k}{y_n + \alpha_n + \lambda_n^k}\sigma_n^k \tag{56}$$

$$h_n^* = -\tfrac{1}{2}\sigma_n^{*2} + \log(y_n + \alpha_n) \tag{57}$$

We can compute $f_n^{k*}$ by simply plugging these values in the definition of Eq. 48.

For a solution to exist, we need $\alpha_n > -y_n$ and $\lambda_n > 0$. This defines the range $\mathcal{S}$.

**Laplace likelihood:** The likelihood is shown in the first equation below, where $b > 0$. The log-likelihood is shown in the second line. We are going to ignore the constant term throughout.

$$p(y_n|\eta_n) = \frac{1}{2b}\exp(-|y_n - \eta_n|/b) \tag{58}$$

$$-\log p(y_n|\eta_n) = |y_n - \eta_n|/b + \text{cnst} \tag{59}$$

We use the following identities to compute expectation wrt a Gaussian $\mathcal{N}(x|h, \sigma^2)$ and its gradient:

$$\mathbb{E}|x| = 2\sigma\phi(h/\sigma) + h[2\Phi(h/\sigma) - 1] \tag{60}$$

$$\nabla_x \Phi(x) = \phi(x) \quad , \quad \nabla_x \phi(x) = -x\phi(x) \tag{61}$$

where $\Phi$ are pdf and cdf function for a standard normal distribution. The last two identities can be obtained easily by simply differentiating the function. The first identity can be proved using the truncated Gaussian moments (see Wikipedia for the expression).

$$\int_0^\infty x\mathcal{N}(x|h, \sigma^2)dx = \sigma\phi(-h/\sigma) + h[1 - \Phi(h/\sigma)] \tag{62}$$

We can express $\mathbb{E}|x|$ as following and then use the above expression to get the first identity. For the last equality, we use the fact that $\phi(x) = \phi(-x)$ and $\Phi(-x) = 1 - \Phi(x)$.

$$\mathbb{E}|x| = \int_0^\infty x\mathcal{N}(x|h, \sigma^2)dx + \int_{-\infty}^0 x\mathcal{N}(x|h, \sigma^2)dx \tag{63}$$

$$= \int_0^\infty x\mathcal{N}(x|h, \sigma^2)dx + \int_0^\infty x\mathcal{N}(x|-h, \sigma^2)dx \tag{64}$$

$$= \sigma\phi(-h/\sigma) + h[1 - \Phi(h/\sigma)] + \sigma\phi(h/\sigma) - h[1 - \Phi(-h/\sigma)] \tag{65}$$

$$= 2\sigma\phi(h/\sigma) + h[2\Phi(h/\sigma) - 1] \tag{66}$$

Define $\eta_n' = (y_n - h_n)/b$, we can do a change of variable and use the first identity to the following expression for $f_n$.

$$f_n(h_n, \sigma_n) := \frac{\sigma_n}{b}[2\phi(h_n') + h_n'\{2\Phi(h_n') - 1\}] \tag{67}$$

where $h_n' = (h_n - y_n)/\sigma_n$.

To compute the gradients wrt $h_n$ and $\sigma_n$, we use Eq. 61.

$$\nabla_{h_n} f_n(h_n, \sigma_n) = \frac{1}{\sigma_n}\nabla_{h_n'} f_n(h_n, \sigma_n) = \frac{1}{b}[2\Phi(h_n') - 1] \tag{68}$$

$$\nabla_{\sigma_n} f_n(h_n, \sigma_n) = \frac{\sigma_n}{b}\nabla_{\sigma_n} h_n' \nabla_{h_n'} f_n(h_n, \sigma_n) + \frac{1}{b}[2\phi(h_n') + h_n'\{2\Phi(h_n') - 1\}] \tag{69}$$

$$= -\frac{1}{b}h_n'[2\Phi(h_n') - 1] + \frac{1}{b}[2\phi(h_n') + h_n'\{2\Phi(h_n') - 1\}] \tag{70}$$

$$= \frac{2}{b}\phi(h_n') \tag{71}$$

Now, we are ready to evaluate the function $f_n^{k*}$. Plugging the into the definition of $f_n^{k*}$, we get the following optimization problem:

$$f_n^{k*}(\alpha_n, \lambda_n) := \max_{h_n, \sigma_n > 0} f_n^k((\alpha_n, \lambda_n, h_n, \sigma_n) \tag{72}$$

$$f_n^k(\alpha_n, \lambda_n, h_n, \sigma_n) := -f_n(h_n, \sigma_n) + \alpha_n h_n + \tfrac{1}{2}\lambda_n \sigma_n^k(2\sigma_n - \sigma_n^k) - \tfrac{1}{2}\lambda_n^k(\sigma_n - \sigma_n^k)^2 \tag{73}$$

Taking derivative wrt to $h_n$ and $\sigma_n$, we get the following:

$$\nabla_{h_n} f_n^k = -\frac{1}{b}[2\Phi(h_n') - 1] + \alpha_n \tag{74}$$

$$\nabla_{\sigma_n} f_n^k = -\frac{2}{b}\phi(h_n') + \lambda_n \sigma_n^k - \lambda_n^k(\sigma_n - \sigma_n^k) \tag{75}$$

Setting the gradient wrt $h_n$ to zero, we get the following:

$$h_n^{'*} = \Phi^{-1}[\tfrac{1}{2}(1 + \alpha_n b)] \tag{76}$$

$$h_n^* = y_n + \sigma_n^* h_n^{'*} \tag{77}$$

Plugging the first equation in the gradient wrt $\sigma_n^*$, setting it to zero, and simplifying as shown below, we get the expression for $\sigma_n^*$.

$$-\frac{2}{b}\phi(h_n^{'*}) + \lambda_n \sigma_n^k - \lambda_n^k(\sigma_n^* - \sigma_n^k) = 0 \tag{78}$$

$$\Rightarrow \quad -\frac{2}{b}\phi(h_n^{'*}) - \lambda_n^k \sigma_n^* + \sigma_n^k(\lambda_n + \lambda_n^k) = 0 \tag{79}$$

$$\Rightarrow \quad \sigma_n^* = \frac{1}{\lambda_n^k}\left[-\frac{2}{b}\phi(h_n^{'*}) + \sigma_n^k(\lambda_n + \lambda_n^k)\right] \tag{80}$$

These results are summarized in the following expressions:

$$h_n^* = y_n + \sigma_n^* h_n^{'*} \tag{81}$$

$$\sigma_n^* = \frac{1}{\lambda_n^k}\left[\sigma_n^k(\lambda_n + \lambda_n^k) - \frac{2}{b}\phi(h_n^{'*})\right] \tag{82}$$

$$h_n^{'*} = \Phi^{-1}[\tfrac{1}{2}(1 + \alpha_n b)] \tag{83}$$

A solution will exist only when $-1 < \alpha_n b < 1$ and $\lambda_n > 0$. Also, it appears that $\sigma_n^*$ can be negative. We can set it to a small positive whenever it is the case.