[Reviews · NeurIPS 2014]

Submitted by Assigned_Reviewer_11

The paper presents new techniques, and theoretical results relating to those techniques, to optimise the Variational Gaussian lower bound to the log evidence in latent Gaussian models LGMs. The technique could be applied to a broader category of latent linear models but their presentation focuses on LGMs only.

VG approximate inference is important because it has many nice properties : it is widely applicable, often quite accurate and relatively fast.

This paper attempts to makes VG methods more scalable without resorting to making factorisation assumptions on the approximating Gaussian distribution.

Whilst the authors show how the objective function can be `decoupled' they do not show experimentally that this leads to a clear improvement in speed or scalability over standard techniques. Their largest problem has only 1.5k data points. I can only imagine that this because there is some scalability issue in their approach/implementation.

Since the covariance of the VG approximation is not explicitly maintained i think it is critical that the authors explain to readers how expectations with respect to this density can be calculated once their solver has found the optimum. We are not just interested in the value of the bound but the moments of the approximating Gaussian also.

The derivations of the theoretical results should be presented in full in the supplementary material. The theoretical results are quite difficult to follow.

Whilst their theorectical contributions are interesting I wonder how useful they are -- solving a lagrangian with non-linear constraints is difficult. Their linearisation method is not guaranteed to converge.

Furthermore, one of the nice things about Seeger's local variational bounding method is that you could specify the rank of the covariance approximation. Could you do that in this context? Or is the entire covariance always approximated -- in which case how scalable is this method anyway.

Quality
The experiments could be more convincing. This paper is about making VG inference scalable and fast? The largest problem to which their method is applied has only 1500 data points -- I wouldn't classify this as large. At this scale they only achieve a marginal improvement over the Cholesky method.

The experiments do not detail the stopping conditions used on either algorithm and so making comparisons is difficult. Also, it is difficult to see what numeric bound values were achieved for either method. I think these results would be better presented in a table, with means and standard errors over multiple runs.

It is especially important that their experiments show the results obtained over multiple runs since their solver is not guaranteed to converge.

Clarity
The authors need to spend a bit more time and effort making the presentation cleaner.

I think the derivations should be presented in full in the supplementary material taking time to describe the notation used and steps taken. As it stands the descriptions of the derivations in the main body of the paper are too short to understand.

Originality
This paper presents novel results about nature of the VG bound optimisation problem. However, the usefulness of these results is unclear -- how fast would the solvers that are guaranteed to converge take in practice? If you only use a solver that isnt guaranteed to converge what is the importance of these results.

Significance
This paper presents some novel results about the VG approximate inference optimisation problem.

Minor points and typos.
line 45 : Maybe Gaussian processes are not the best example of models with strong correlations between variables in this context since we would only attempt approximate inference for GP models when O(D^2) space and O(D^3) time is feasible and so wouldnt need a distributed solver.

* Section 2 para 1. I appreciate that it is convenient to focus on just LGMs in this paper but i think readers would appreciate a more precise description of the broader class of models that your method could be applied to.

Eq. (5) maybe explain the log det V notation.

line 104: expand with no s

line 134: it doesnt make sense to introduce auxiliary variables such that h_n = \bar{m_n}. What are the constraints for then? Either you introduce the auxiliary variables and the constraints or you just set h_n to be defined as \bar{m}_n.

line 141: concave and convex mix up? should it not say ...function f (concave with respect to Cholesky of V).

eq(6) Maybe say subject to constraints ... for n=1,..,N.

Theorem 4.1 eq 8 -- you should explain that \Lambda^* = \diag(\lambda^*) .

line 159. missing n subscript on \lambda^*

eq (9) : i think notation used for the partial derivatives is a little unclear. You either need to explain the notation used or write the partial derivate out explicitly with a vertical "evaluated at" bar with subscript h=h^*

line 169 : the condintions that

paragraph 1 page 4 : maybe change each of the "The x part..." to "Part x ..."

line 177 : constrain ",
line 177 : this sentence is weird ", and similar to [9]"

line 179: "while the second equality is simply change of variables"

line 190: there are two prerequisites...

lines 194 to 202 : This is hard to follow. I would recommend taking your time, going through each of the steps explicitly and precisely defining terms in the supplementary material.

line 211 : this sentence is poor.

line 219 : "this optimisation appears..." which optimisation? Both formulations are difficult?

line 229 : "and is an effective for " method?

line 254 : "for primal function" for the primal function?

line 260 : missing super script k on f ?

line 266 : of the dual variables are equal

line 285 : two eithers.

line 291 : see Algorithm 1?

line 293 : quadratic convergence under what conditions. The next sentence is missing a "this" it is also unclear.

line 294: we never observe this in

line 312 : Sentence starting "Basically, ..." What is the justification for this heuristic?

Bibliography : Authors in references need to be made consistent
Summary: Present theoretical results that allow the VG approximate inference objective function to be optimised in a decoupled fashion. Experiments do not present a convincing improvement in speed or scalability.

Submitted by Assigned_Reviewer_31

Decoupled Variational Gaussian inference

The paper looks at the widely-used variational Gaussian approach and ways in which inference in these settings can be accelerated. There are two main contributions: the paper brings together under the unified theory of dual decomposition the observations and reparameterisations that have been used in efficient variational Gaussian methods. Secondly, the paper describes how the optimisation in the dual functions obtained can be done efficiently. The paper described these very clearly and the main theorem and all the decompositions are easy to follow and re-verify.

The notation is designed to be the most general to unify with many other latent Gaussian models, but I think this makes it difficult to easily communicate this. In particular making the distinction of what the indices on observations and dimensions should be looked at again. I read the paper with Gaussian latent variable models in mind, where N will be the dimensions of a single data observation, whereas N will be an index over observations if we are in the regression setting. Many might not see this and affects the impact the paper could have. To help here, it might be useful to include the graphical model, and/or restrict to a particular model class (regression, LVM, kernel-based, etc).

The core part of the paper was section 5, and more insight and guidance would be useful here. What is the insight into the use of the linearly constrained Lagrangian, what are it's advantages and disadvantages, when do we expect it to work well.
The notion of parallelisation was not very clear in the description and this can be looked at again.

Overall, the paper addresses questions of inference, but not parameter learning (if we were in latent variable models). The M-step equations could be very expensive, and despite the gain from this smarter inference, in the LV setting we would overall have not made any gains in terms of computational cost. Some explicit discussion on complexity would make this point and discussion around approaches for further gains in efficiency. The paper raises many questions as to what approaches ultimately provide substantive reductions in computation, but this is part of the ongoing work in this area.

The experiments are still small-scale and we would need to start addressing larger-scale problems. This line of research has deviated from the large body of variational research now using stochastic approximation methods and other Monte Carlo approaches, and might be useful to begin to make these connections.

Summary: I enjoyed reading this paper. All the equations are very easy to follow and re-derive and the insight in to dual methods and augmented Lagragian algorithms is useful to have. Experiments remain small-scale and restricted to GP regression, but allude to useful benefits in many parts of probabilistic modelling.

Submitted by Assigned_Reviewer_40

Overview

The authors proposed variational Gaussian inference method that optimize a lower bound of marginal likelihood of Latent Gaussian Models (LGMs). LGMs are popular models, which are typically used for tasks like Gaussian Process Regression/Classification. The authors decouple the mean and covariance of the variational posterior distribution in the likelihood part of marginal likelihood from the mean and covariance in the KL-divergence part, and apply a Linearly Constrained Lagrangian (LCL) method for one of the resulting constraints, and formulate the inference by its dual problem.
Quality

The decoupling of variational parameters enables an efficient parameterization with LCL. This parameterization avoids the optimization of the covariance matrix of the variational posterior distribution, which is problematic due to its positive-definite constraint. In the new parameterization, optimization formulates as a sequence of 2D concave optimization, which is more efficient and significantly reduce the number of parameters. Moreover, by adding an additional Lagrangian term, the convergence of not log-concave noise model can be improved.

The idea is decoupling variational parameters with LCL is new and very interesting. I have several reservations regarding the proposed method. The claimed efficiency of parameterization is due to the dual formulation, which optimize the mean and variance of the noise model instead of the mean and covariance of the variational posterior distribution. However, as far as I understand, if the dimensionality of output is the same as the dimensionality of the latent space, then the dual formulation would not be more efficient.

For the case of not log-concave noise model, the proposed model makes use of augmented Lagrangian methods for improving converge. It is known that LCL approaches might not be reliable from different starting points (initial parameters). It would be very interesting at least by experiments to show the reliability of the proposed method with a not log-concave noise model.

Clarity

This paper is well written, and easy to follow. The result section needs to be improved with more experiments with different noise model.

Originality

This work looks like an original work with an interesting idea and good performance.

Significance

This work presents an interesting approach of variational inference for non-Gaussian noise model. It could potentially be very useful for Gaussian Process related models.
Summary: This work presents an interesting approach of variational inference for non-Gaussian noise model by decoupling variational parameters. The method is well explained. The paper can be further improved with additional experiments demonstrating convergence and reliability.
Author Feedback
Author rebuttal: We would like to thank all the reviewers for their comments.

**** Two main points that were unclear in the paper ****
(1) An important motivation of this work is to enable efficient implementation of a non-conjugate model using the implementation of a conjugate model. For example, to implement Bayesian logistic regression, LCL method only requires an efficient implementation of Bayesian linear regression (see Eq. 18). Similarly, GP classification requires an efficient implementation of GP regression only. In fact, if we want to do sparse GP classification, it is sufficient to have an efficient code for sparse GP regression. In Algorithm 1.1, line 275 corresponds to these conjugate-model computations. All the other computations can be done in O(N), parallely. We will modify Section 5.1 to include this discussion (around Eq. 18 in the paper).

(2) Another important motivation is to make VG approximation possible for large-scale data. For large problems, where covariance matrix is bigger than 10k x 10k, Cholesky method cannot be run unless we make extra factorization assumptions. Our method makes this possible by reducing the number of parameters, while maintaining a good convergence rate similar to the Cholesky method.

**** Specific comments about the Results section ****
* First, we would like to apologize for a small bug in our code which initialized LCL at a bad position (initialization of lambda was done at exp(1) instead of exp(0) ). After correction, we see a significant improvements in the plot.
* As reviewer 11 suggested, there are some issues with our current implementation. These issues, however, are not due to any fundamental flaws in the algorithm, rather are due to our implementation. Our current implementation is done in Matlab and therefore hides the improvements obtained due to parallelization, specially for large data. To correct this, we are working towards implementation in a well-known open-source machine learning toolbox written in C++.
* We also agree that the results section is not extensive. We did not have all the results until the deadline. See the list below for some additional results that we will include in the paper.

**** We will make the following changes in the Results section *****
* We will correct the bug in Fig. 1 and include the corrected version.
* We will include a large-scale result for Bayesian logistic regression (on real data) where number of features (L) are larger than the number of samples (N). We will make sure that L > 10000, ensuring that the example is large enough.
* We will include results for multiple runs showing global convergence. We will extend the discussion of Section 4.2 illustrating schemes for global convergence. We will also include a table showing the final values of the objective function and time taken for convergence.
* We will include an experiment for non-concave case for GP regression with T-distribution.

***** Comments for Reviewer 11 *****
Following your comments, we will also include the following.
* details about computing expectation with the posteriors efficiently.
* an elaborate proof of the theorem in the Appendix.
* a discussion about extensions to general (continuous) latent variable models.

We agree with your suggestions for the typos and other correction and we will include those.

Line 312 is not a heuristic, but is a theoretical fact. Please see the details Chapter 4.2 of Bertsekas book (reference [1]). These points are important for global convergence of the method, which is an important worry, as you point out. There are enough tools in the optimization literature which allow us to make these methods globally convergent (e.g. see [4]). We will include a discussion of these in the paper.

You can easily make a low-rank approximation by choosing only few lambda (and setting rest to zero). This also corresponds to only including few constraints in the optimization problem of Eq. 6. You can ensure convergence by resorting to stochastic methods (see comments to Reviewer 31) which we leave as future work. Our algorithm is very similar to Seeger's method, but we do not make a lower bounding assumption on the likelihood terms.

**** Comments for Reviewer 31 ****
Including a graphical model is a very good idea. We will also add few examples to avoid confusion about the notation. Regarding your question about stopping criteria, all algorithms are stopped when the lower bound values of Eq (5) are less than 1e-4.

You comment about parameter learning and stochastic method is an important one and is missing from the paper. Thanks! We will include the following discussion in future work section. Existing methods on stochastic variational inference (SVI) use mean-field and are widely applied for conjugate models (such as LDA). We believe that our method is a stepping stone towards generalizing SVI methods to non-conjugate models without having to resort to mean-field approximations. The idea is to stochastically include only few constraints to maximize Eq. (6), which amounts to a low-rank approximation of the covariance matrix. This can be used to construct an unbiased estimate of the gradient, to be used in the M-step to update the ``global" variables. We will investigate this in the future.

**** Comments for Reviewer 41 *****
As discussed earlier, we are going to include a few new results which hopefully answers your questions. Here is a clarification to the following question you asked: “If the dimensionality of output is the same as input, dual formulation will not be efficient”. Actually, if the dimensionality of output is much much more than square of the dimensionality of input, then it is better to work with the primal. In our notation, if N >> L^2, then we should use primal. This is not the case for most models, e.g. for GP, L=N, so primal is inefficient. For classification problems, where number of features (L) is of the order of N, it is better to work with the dual.